# Investigating the Market Value of Brumbies (*Equus caballus*) in the Australian Riding Horse Market

**DOI:** 10.3390/ani13091481

**Published:** 2023-04-27

**Authors:** Victoria Condon, Bethany Wilson, Peter J. S. Fleming, Brooke P. A. Kennedy, Tamara Keeley, Jamie Barwick, Paul McGreevy

**Affiliations:** 1School of Environment and Rural Science, University of New England, Armidale, NSW 2350, Australia; 2School of Life and Environmental Science, Faculty of Science, University of Sydney, Camperdown, NSW 2006, Australia; 3Vertebrate Pest Research Unit, NSW Department of Primary Industries, Orange Agricultural Institute, 1447 Forest Road, Orange, NSW 2800, Australia; 4Ecosystem Management, University of New England, Armidale, NSW 2351, Australia; 5School of Agriculture and Food Sciences, University of Queensland, Gatton, QLD 4343, Australia; 6One Welfare Research Institute, Faculty of Science, Agriculture, Business and Law, University of New England, Armidale, NSW 2350, Australia

**Keywords:** brumby, feral horse, horse sale market, One Welfare, recreational riding, wild horse

## Abstract

**Simple Summary:**

Feral horse (*Equus caballus*) numbers in Australian national parks are increasing and population management typically requires culling to reduce numbers and deleterious impacts. Alternatives to culling include capture either for permanent removal or relocation, the application of fertility control interventions or rehoming. All these have been proposed for feral horse management globally, and all are subject to challenges, costs, and potential welfare threats. Rehoming as a management tool has wide community acceptance but, to date, there has been no research to determine the number and value of feral horses (brumbies) entering the Australian recreational riding horse market following rehoming. We analysed data from Australia’s leading monthly horse-trading magazine *Horse Deals*, between February 2017 to July 2022 to determine which factors influenced the estimated market value of rehomed Australian brumbies. Rehomed Australian feral horses were priced differently to similar domestic-bred horses, and most were described as “Unbroken”. Further research is required to determine what niche feral horses occupy in the recreational riding horse market and which rider demographic is best suited to these horses to optimise horse-human welfare outcomes.

**Abstract:**

Feral horses, also known as brumbies, are widely distributed across Australia with some populations being managed largely by human intervention. Rehoming of suitable feral horses following passive trapping has wide community acceptance as a management tool. However, there is little information about the number and relative economic value of feral horses compared with cohorts in the riding horse market. We examined 15,404 advertisements of horses for sale in 53 editions of *Horse Deals*, published from February 2017 to July 2022. Despite the considerable media attention and public scrutiny surrounding feral horse management, rehomed feral horses represented only a tiny fraction of the horse market in the current study. Of the 15,404 advertisements examined, only 128 (0.0083%) were for feral horses. We recorded phrases used to describe behavioural characteristics and other variables. The following variables were found to be not independent: Ridden Status, Height, Age, Sex, Colour, and Warning terms/more work. Using descriptive statistics to describe basic features of the data, the average price for feral horses ($1408) was lower than that for domestic horses ($1790) with the maximum price for a domestic horse being nearly twice the maximum for a feral horse. Univariate analysis showed feral horses were over-represented among “Unbroken” horses and underrepresented among “Ridden”, “Broodmare” and “Harness” horses compared with domestic bred horses (*p* < 0.001). Feral horses appeared over-represented at shorter heights, among younger age groups (3 years or younger and 3.1 to 6 years) (*p* < 0.001) and in the dilute colour category (*p* = 0.008). The multivariable mixed model on price revealed that for domestic horses, the highest estimated marginal mean price averaged across the colour categories was for ridden horses aged 6.1–10-year-old at $1657.04 (95% CI $1320.56–$2074.66). In contrast, for feral horses, the multivariable mixed model demonstrated the similar highest estimated marginal mean averaged was for green broken 3–6-year-old horses that have undergone foundation training under saddle at $2526.97 (95% CI $1505.63–$4208.27). Australian feral horses were valued differently tfromsimilar domestic horses in the recreational riding horse market and further research is warranted to determine appropriate target markets and boost the sustainability of rehoming as a feral horse management tool.

## 1. Introduction

Few animals evoke as much passion, diverse public opinion and romantic ideology as free-roaming or wild horses (*Equus caballus*). The management of free-roaming horses, known as mustangs in the United States of America, wild horses in New Zealand (including the well-studied Kaimanawa population) and brumbies in Australia, is typically complex and multi-faceted. Management may be affected by the usage of horses as livestock and companion animals, their aesthetic appeal, and social constructs that regard free-roaming horses as heritage species, cultural totems, wildlife or invasive pests [1]. Successful free-roaming horse management requires an understanding of the relative value of the horses to stakeholders with opposing perspectives [2]. That is, an animal regarded as a pest species by one stakeholder may be prized as a companion animal by another. The rehoming of brumbies as a management tool has broad social appeal, and there are numerous anecdotal examples of brumbies being successfully rehomed following their removal from national parks [3]. However, despite community support, brumby rehoming opportunities are often undersubscribed, and this lack of demand has been identified as a bottleneck in the successful implementation of rehoming as a management tool [4].

Brumbies may enter the recreational horse market following capture and subsequent rehoming. Rehomed brumbies may then be on-sold through sale yards, by private sale, or through one of the dedicated brumby rehoming organisations in Australia. Currently, there are no publicly available, objective data about the number and value of brumbies being offered for sale in the Australian recreational riding horse market through any of these mediums. This lack of reliable information may lead to unsupported, misinformed assumptions about the value of rehomed brumbies compared with other cohort groups. Unlike an auction forum, public sale yard or use of an intermediary group, private selling by advertisement requires the vendor to estimate the market value of their horse. When valuing horses for sale, research has shown that vendors and purchasers value different characteristics [5,6]. For example, vendors of Irish Sport Horses rated sex, colour, experience, and performance as important attributes, whereas potential purchasers placed more value on temperament and aesthetic appeal [6]. Relative value may also differ according to breed [7]. It has previously been reported that covert warning (negative) descriptors in pony and recreational (non-thoroughbred) riding horse advertisements had a negative influence on asking price whilst, conversely, positive descriptors associated with handler and/or rider safety had a neutral and less significant effect on asking price respectively in these two cohorts [5,8]. In contrast to these findings, positive descriptors were found to have a significant positive effect on the asking price of Thoroughbreds in the Australian recreational riding horse market [7], demonstrating a breed difference.

Poor compatibility between horse and rider, known as a “mismatch” [9], can compromise safety and may also compromise both horse and human welfare. The equestrian industry has traditionally focussed on the use of personal protective equipment (PPE) such as helmets, boots, gloves and body protectors to minimise the consequences of equestrian-related injuries [10]. However, it has been suggested that PPE alone is insufficient as a sole risk-management strategy [10]. Higher-level safety controls to further mitigate risk include appropriate rider–horse matching and understanding which horses (age, breed, experience, education and history) are most likely to be unpredictable, and therefore less suitable, for certain riders [10,11]. For these reasons, it is important to understand what niche feral horses occupy in the recreational riding horse market to optimise horse-human welfare outcomes. The current study investigates variates associated with the vendor advertised price of horses and does not identify purchaser demographics, but some insights can be gained by initially determining what qualities are valued by the vendors of feral horses with a view to understanding their target market.

The Australian brumby currently has pluralistic status as both a culturally significant icon, recognisable in Australian colonial folklore, and an introduced pest [12]. However, a review of the grey literature demonstrates the early settlers did not share the modern appreciation of the brumby as either an Australian icon or a valued riding horse. Newspapers and journals throughout the 19th century repeatedly demonstrated hostile attitudes toward brumbies [13,14,15]. There was no room for sentimentalism where the brumby conflicted with the economic interests of pastoralists, eating “his grass or interfering with his stock horses” [15] and there is no evidence of romantic notions about brumbies. Welfare concerns were also not expressed, and the brutal reality and economics of a colonial society underpinned such a pragmatic approach. This contrasts with a growing regard for animal welfare in the UK at about the same time and the earlier publication of *Black Beauty* [16] that took the, then unusual, step of anthropomorphising a horse as he narrates his biography and reflects on his treatment at the hands of humans. However, following World War II, Australian brumbies were popularised through the fictional series of *The Silver Brumby* novels [17]. These works, along with the iconography of *The Man from Snowy River* [18], contributed to the romanticisation of brumbies. Meanwhile, perhaps because fences keep most stray horses out of agricultural land the Australian pastoralists’ earlier antagonistic and utilitarian views of free-roaming horses have been largely relegated to history.

Currently, the control of free-living horse populations, including brumbies, is a contentious and complex issue worldwide [19,20,21]. Horses are often regarded as charismatic animals, and in the global debates surrounding feral animal population control, free-roaming horses frequently appear to be the exception [22,23,24,25]. Indeed, one study even suggested that in the USA, free-roaming horses were “valued above all other animals in our society” [23]. The social costs and benefits of wild and free-living animals such as horses, including nonmonetary existence values [26], have been the subject of much discussion [20,27,28,29]. Whilst the societal and cultural value of free-living horses may be unquantifiable, the economic value of brumbies in the Australian riding horse market has to date not been scrutinized. Accurate, objective data about the economic value of brumbies relative to other cohort groups is important to inform public debate and potentially provide insight into the viability of brumby rehoming programs.

We note various terminology such as “feral horse”, “wild horse” and the colloquial “brumby” have been used to describe free-roaming horses in Australia. This nomenclature can be a source of controversy. Przewalski’s horse (*Equus ferus przewalskii*) is the only extant true wild horse in the world [30] and all free-ranging horses globally have a domestic origin despite their various backgrounds [31]. Horses are an introduced rather than indigenous species in Australia [32] and the current study used the terms “feral” and “domestic”, based on the Cambridge Dictionary definitions, to describe the two cohorts. [33]. That is, a “domestic” animal is one that is not wild and is kept as a pet or to produce food and a “feral” species is one that has become wild from a state of cultivation, domestication or selection by humans [34], which is the appropriate descriptor for Australia’s free-roaming horses. However, the colloquial term “brumby” was also used when searching and referring to the *Horse Deals* photo advertisements used for data collection. “Brumby” is the descriptive term commonly used by feral horse vendors in the context othe f type of horse for sale, furthermore no horses in the 15,404 photo advertisements examined were breed-described as “feral” or “wild” horses.

The aim of our study was to determine factors influencing the economic value of rehomed Australian brumbies. We analysed the details of photo advertisements placed in the Australian leading monthly horse-trading magazine *Horse Deals* to examine variates associated with pricing in the Australian recreational riding horse market. These data were used to investigate the hypothesis that brumbies and comparable domestic-bred horses are valued differently from other cohort groups in the Australian market. 

## 2. Materials and Methods

### 2.1. Data Collection

*Horse Deals* magazine (Agricultural Press Pty Ltd., Mount Gambier, South Australia) is a monthly Australian publication with a magazine readership of 60,000 per month [35]. It provides a print and online forum for owners and trainers to advertise horses for sale. The magazine and associated website reach an estimated audience of 210,000 per month [36]. This platform was chosen as it attracts buyers and sellers of equids from a diverse range of breeds and disciplines and has been used to provide data in previous studies [5,7,8].

The magazine does not include a specific section for horses identified as brumbies. Advice from the magazine editor indicated that advertised brumbies would likely be included in one of three sections: “Allrounders”, “Coloured” and “$1000 and Under”. We examined 15,404 photo advertisements of horses for sale in these sections in 53 editions of *Horse Deals*, published from February 2017 to July 2022: 6902 from the “Allrounders” section, 2383 from “Coloured” and 6119 from “$1000 and Under”. Issues of Horse Deals magazine used to compile the dataset are shown in Table 1.

Advertisements for brumbies did not appear in every issue. Data for brumby advertisements which appeared in more than one issue (*n* = 9) were recorded the first time the horse was advertised.

### 2.2. Inclusion Criteria

From the total pool of 15,404 photo advertisements, we gathered data for feral horses from 128 advertisements. The details of an advertisement were included only if the following breed descriptors were used: brumby (*n* = 111), Guy Fawkes (horses captured from Guy Fawkes National Park in northern NSW) (*n* = 15), heritage (horse) (*n* = 2). Animals described as “brumby cross” were excluded. Advertisements not listing a specified price, for example: “price on application”, “for tender,” or “for lease” were also excluded. We assumed that the domestic horses advertised adjacently to a brumby were randomly placed with reference to the brumby and we used these (*n* = 256) for comparison of the same details in feral horse advertisements. 

### 2.3. Identification of Phrases Used to Describe Behavioural Characteristics

We identified 296 phrases that vendors used to describe the behavioural characteristics of the advertised horses. The descriptors set out in Table 2 were assigned to one of four categories (*very reassuring, somewhat reassuring, neutral,* and *warning/more work*) according to the perceived level of reassurance they offered about behavioural characteristics associated with safety for the rider/handler. The four categories were the same as those used by Hawson et al. (2011) [5]. Significant additions were made to Hawson *et al*.*’s* list of descriptors, chiefly to reflect terms more relevant to the equids of interest rather than only to ponies. As previously published [5,7,8], three of these categories (*very reassuring, somewhat reassuring, neutral*) reflected degrees of potential positive reassurance of behavioural characteristics associated with the safety of the rider/handler, whereas the fourth (*warning/more work*) reflected covert warning (negative) descriptors. In the current mixed population, the fourth category combined warning descriptors with phrases indicating the need for “more work” to be conducted before the horse could be safely used for recreational riding. As the study examined variates associated with pricing in the Australian recreational riding horse market, phrases were categorised to reflect the horses’ potential to be used as a recreational riding horses. The same behavioural characteristics filter was applied to both cohorts (brumbies and domestic horses).

Behavioural descriptors were given a score of +1 for each mention of behaviour that corresponded with one of the four categories described. Each category was scored the same i.e., no post-hoc weighting was applied. Behavioural descriptors found in advertisements for feral and domestic horses were allocated to a category based on the level of reassurance inferred by the statements (see Table 2). Three footnotes were added: * Aspirational statement, † More work required before riding as a recreational mount and ‡ Specific to brumbies. “Aspirational statements” describe the vendors’ future expectations of the horse rather than the horses’ current status and behaviour, e.g., “make good campdraft or mustering horse”, “make good horse in the right hands” and “great potential”. “More work required before riding as a recreational mount” is self-explanatory and other similar statements included “Needs a lot of work”, “Unhandled” and “Needs finishing”. Such horses cannot be conclusively characterised as recreational riding horses without a further investment of time and/or money to advance their training. Other statements such as “very quiet for a brumby”, “domesticated for over a year” and “captured/trapped as part of capturing program” were noted as statements specific to feral horses only.

### 2.4. Other Variables Recorded

In addition to behavioural descriptors, we recorded the following data: page number, category (Allrounders, Coloured, $1000 and Under), size of advertisement (half page, quarter page, one-eighth page, one-sixteenth page), price (AUD), height (in hands: one hand is equivalent to 10.2 cm), age (years), sex (mare, filly, gelding, stallion, colt, not stated), colour (black, brown, chestnut, bay, grey, dilute, coloured), ridden status (ridden, unbroken, green broken, unknown, broodmare, harness), location (state (Queensland, New South Wales, Victoria, South Australia, Western Australia, Tasmania, Australian Capital Territory or Northern Territory) and town) and registration status (registered or not registered with any Australian brumby society) to investigate variates associated with pricing in the Australian recreational riding horse market. 

Horses advertised as green broken have undergone initial foundation training under saddle (breaking-in) but have had minimal further training. A green-broken horse is considered “unfinished”, introduced to a saddle and with some basic ridden experience [37]. The term is broadly understood in the equestrian industry. However, because the phrase is subjective, green broken horses may have had either days or weeks of training, but not enough to be deemed a finished, fully-trained, experienced riding horse [11]. 

All advertisements in this study referred to height as “hands high”, which is abbreviated to “hh”. Height was converted to a categorical measure based on full hands. Height categories then became: less than 14 hh, 14 to less than 15.0 hh, 15 to less than 16.0 hh, 16.0 or more. Similar to the height variable, age was treated as a categorical value without rank due to the presence of “weanlings” and suspected non-random missing values. Advertised horses with any of the coat colours recognised by the Dilutes Australia studbook (buckskin, smoky black, champagne, cremello, perlino, smoky cream, dun, mushroom, palomino, pearl, silver or non-solid dilute) [38] were pooled into a single “Dilute” category for colour. Advertisements may have also made mention of other characteristics including, conformation descriptors, rider experience and vices/stereotypies (stylized, repetitive, apparently functionless motor responses [39]). These variates were recorded, but as each was present in only a small fraction of the advertisements, they were not suitable for further analysis.

### 2.5. Statistical Analysis

Descriptive statistics were used to quantitatively describe the basic features of the data. Advertised price was the dependent variable of interest in this study. During preliminary analysis, Chi-square and/or Fisher’s exact tests were run on explanatory variables using R [40]. Price was positively skewed (Figure 1A), therefore was less than likely to meet the assumption of normally distributed residuals in linear regression. Transformation of the data to the log normal scale was complicated by the presence of zeroes. As the number of zeroes was relatively small (approximately 1.8%), a small constant ($75: half the lowest non-zero value) was added to the log price consistent with previous literature [41] to account for the undefined nature of log(0) (Figure 1B) [42]. All regression computations were performed using the transformed price data: log(Price(AUD) + 75).

A multivariable regression between price and explanatory variables was used to identify variables that were associated with log(price(AUD) + 75) for pooled domestic and feral advertisements.

The explanatory variables considered for the regression were: Ridden Status, Height, Age, Sex, Colour, Very Reassuring terms, Somewhat Reassuring terms, Neutral terms, Warning terms/more work and Advertisement Size. Random variables identified were: Magazine Issues and page nested in the Year of Advertisement. These explanatory variables were additively modelled as fixed and random effects using R packages lme4 [43] and lmer test [44], proceeding through a stepwise deletion until all remaining terms were *p* < 0.20 or until Akaike information criterion (AIC) and Bayesian information criterion (BIC) began to rise.

Interaction terms among the fixed effects were trialled by adding them to the resulting reduced model. Those with *p* < 0.25 (Table 3) were then all simultaneously added to the reduced model and a second stepwise deletion using the same criteria as above was performed. The final model included Feral status, ridden status, age, colour, very reassuring descriptors, somewhat reassuring descriptors and warnings as fixed effects. Interaction terms in the final model included Feral status: Ridden status, Feral status: warnings, ridden status: age, ridden status: colour, and age: somewhat reassuring predictors. Compared with the reduced additive (fixed) mixed model, a model with added interaction terms explained significantly more deviance (chi = 105.76, *df* = 46, *p* < 0.001). Residual and QQ plots were inspected for normality and heteroscedasticity and judged acceptably. 

## 3. Results

The following variables were not independent: Ridden Status, Height, Age, Sex, Colour and Warning terms/more work. The counts among domestic and feral horses for explanatory variables reaching significance are shown in Table 4 with expected values from the null hypothesis in brackets.

### 3.1. Price

Domestic horses included in the sample represented a total perceived value of approximately AUD 458,259 whereas the total perceived value for feral horses was approximately AUD 180,199. The collective total perceived value of the sample was AUD 638,458 and the median overall price for the 384 horses was $1000, while the mean was $1663, SD = $1490.26. Domestic and feral horses had the same median price ($1000), while the average price for domestic horses ($1790, SD = $1622.44) was higher than that for feral horses ($1408, SD = $1146.30). The maximum price for a domestic horse was nearly twice the maximum for a feral horse (Figure 1A). 

### 3.2. Ridden Status

Univariate analysis showed feral horses were over-represented among “unbroken” horses and underrepresented among “ridden”, “broodmare” and “harness” horses compared with domestic-bred horses (Table 4). When horses were grouped by “ridden status”, feral horses had an overall lower status than domestic horses (Table 4, Figure 2). The ridden category consisted of 55.08% (*n* = 141) domestic and 26.56% (*n* = 34) feral horses. Correspondingly, 23.05% (*n* = 59) of domestic and 54.69% (*n* = 70) feral horses were advertised as unbroken. Fewer domestic horses (5.47%, *n* = 14) were in the green broken category than feral horses (17.19%, *n* = 22). There were no advertised feral horses in the broodmare or harness categories. 

### 3.3. Height

Feral horses were over-represented at shorter heights and under-represented at greater heights. Domestic horse heights appeared more evenly distributed than feral horse heights: 23.8% (*n* = 61) of domestic horses were less than 14 hh, 20.3% (*n* = 52) measured 14 to less than 15 hh, 34.0% (*n* = 87) were 15 to less than 16 hh and 21.1% (*n* = 54) were 16 hh or higher. No height was listed for 0.8% (*n* = 2) of domestic horses. In contrast, most feral horses were under 15 hh with 41.4% (*n* = 53) measuring less than 14 hh and 49.2% (*n* = 63) in the 14 to less than 15 hh category. Only 7.8% (*n* = 10) of feral horses in the sample were advertised as over 15 hh and 1.6% (*n* = 2) had no height listed. 

### 3.4. Age

Feral horses were over-represented among younger age groups (3 years or younger and 3.1 to 6 years). Overall, 43% (*n* = 110) of domestic horses were aged under 6 years, in contrast with most feral horses (77.3%; *n* = 99). Fewer (19.1%; *n* = 49) domestic horses were aged 3 years or younger, compared with 42.2% (*n* = 54) of feral horses. Horses aged between 3.1 to 6 years accounted for 23.8% (*n* = 61) of the domestic horse sample and 35.2% (*n* = 45) of feral horses. The 6.1 to less than 10 years age category included 27.3% (*n* = 70) of domestic horses and 17.2% (*n* = 22) of the feral horse sample. Only 4.7% (*n* = 6) of feral horses were aged over 10 years, compared with 28.1% (*n* = 72) domestic horses. Age was not listed in 1.6% (*n* = 4) domestic horse and 0.08% (*n* = 1) feral horse advertisements. 

### 3.5. Colour

Colour was also not independent of being feral or domestic (*p* = 0.008). The most important contributors to the difference between observed and expected values in the colour variable were the under-representation of coloured horses and over-representation of dilutes among feral horses. In advertised domestic horses, bay (33.6%; *n* = 86) was the most common colour, followed by chestnut (21.1%; *n* = 54), coloured (18.4%; *n* = 47), brown (7.8%; *n* = 20), grey (7.0%; *n* = 18) and dilute (6.6%, *n* = 17) with black (5.5%; *n* = 14) the least commonly advertised. Bay (39.8%; *n* = 51) was also the most common colour among feral horses followed by chestnut (25%; *n* = 32), dilute (12.5%; *n* = 16) and brown (8.6%; *n* = 11). Grey, coloured and black feral horses were all equally least common (4.9%; *n* = 6).

### 3.6. Statistical Analysis

#### Multivariable Mixed Model on Price

Results of the multivariable regression between price and explanatory variables are summarised in a type three ANOVA table [45] (Table 5).

For example, the interaction [46] between age and ridden_status for each of the groups (averaged over the levels of colour) can be seen in Figure 3 [42].

To describe these data, we note that the effect of ridden status varied significantly overall by group (χ^2^ = 14.31, *df* = 3, *p* = 0.003) but not quite by age (χ^2^ = 26.04, *df* = 16, *p* = 0.053). The effect of ridden status also varied significantly by colour (χ^2^ = 51.38, *df* = 23, *p* = 0.001). Overall, feral horses were advertised for slightly less than domestic horses (χ^2^ = 5.64, *df* = 1, *p* = 0.018) but this pattern did not hold for green broken horses. Green broken horses were less expensive than unbroken horses among domestic breeds (e.g., about 50% less for 3-year-olds). Conversely, feral horses that were green broken attracted a higher advertised price (e.g., about 50% more for a less than 3-year-old). Indeed, the highest estimated marginal mean price when averaged across the colour categories was for green broken 3–6 year-old feral horses at $2526.97 (95% CI $1505.63–$4208.27).

Increased use of *warning/more work* terms in the advertisement was significantly associated with lower prices among domestic horses (*t* = −2.701, *p* = 0.007). That trend was absent in feral horses (χ^2^ = 1.47, *df* = 1, *p* = 0.225, Figure 4). 

## 4. Discussion

The aim of this study was to investigate variates associated with the advertised price of horses in the Australian recreational riding horse market. To the authors’ knowledge, our study is the first analysis of feral horses entering the riding horse market. Australian feral horses (brumbies) were valued differently from similar domestic-bred horses. We acknowledge that rehomed feral horses sold through various media and print advertisements, such as those in *Horse Deals* magazine, do not necessarily represent the marketing of the entire population. Nevertheless, the use of data from this forum provides valuable insights into the economic value of feral horses relative to other cohort groups advertised in the same medium. 

The average advertised price for domestic horses ($1790) was higher than that for feral horses ($1408) and the maximum price for a domestic horse was nearly twice the maximum for a feral horse. While prices for feral horses were comparatively low overall, green broken feral horses, i.e., those that are not experienced or extensively trained, were a notable exception. Chapman et al. (2020) defined an advanced rider as “able to ride a green-broke-young horse” [11]. Green-broken domestic horses were less expensive than their unbroken counterparts but for feral horses, being green broken resulted in an increase in price (Figure 3). The interaction plots demonstrating the effect of age and ridden status for both cohorts (Figure 3) show an inverted pattern. For domestic horses, the highest estimated marginal mean price averaged across the colour categories was for ridden horses aged 6.1–10 year-old at $1657.04 (95% CI $1320.56–$2074.66). A ridden horse of this age may reflect a level of experience which is valued positively by prospective purchasers. This observation is in accordance with the results of studies into rider preferences [47,48,49]. For example, a study of the horse-buyers’ market by Gille et al. (2010) revealed 47.5% of respondents identified as “amateurs” [48]. When purchasing a horse, this amateur group considered training level as the most important criterion and assigned a high value to the ease of handling and riding. Graf et al. (2013) found character, rideability, willingness to work, and temperament were important to leisure riders [49]. Accordingly, in an “Ideal Horse Questionnaire” [47] “behaviour during riding and handling” was selected as the most important quality by 43.6% of respondents. Such findings indicate that rideability is valued by riders and those who ride for pleasure appear to have little interest in riding a challenging or naïve/untrained horse. A horse with more experience may also represent an investment of money, training and/or time by the vendor, justifying an increased price [7,8]. 

Compared with domestic horses, the differences in interactions amongst advertised price, age and ridden status in the feral horse cohort were distinct. Hennessy et al. (2008) noted a discrepancy in the importance attached to equine characteristics by vendors compared with purchasers [6] and, of course, all of these prices have been set by vendors. This raises the question: why do feral horse vendors consider relatively young (less than 10 years) green broken feral horses to be more valuable than their ridden equivalents? Given that previous research demonstrates that leisure riders value rideability and ease of handling, the vendor’s target market for feral horses appears unclear. Additionally, with the estimated cost of professional horse training currently ranging between $400-$650 per week, depending on requirements, the value of such training to improve rideability in this low price bracket seems questionable.

Clearly, not every recreational rider has the same preferences. Górecka-Bruzda et al. (2011) found a high level of risk and challenge was valued by some groups of riders [47]. Horseback riding is known to be a dangerous activity and injury can occur from merely handling or being in contact with horses [10,50]. The risk of hospital admission from horseback riding is higher than from football, rugby, skiing and motorcycle racing [10,51]. It seems plausible therefore that riding a more challenging horse may equate to a higher level of risk with more possibility for injury. Chapman et al. (2020) found risk acceptance was common among horse riders and danger was even seen as an inherent part of human–horse interactions [11]. Chapman et al. also suggested that opinions of participants in equestrian activities tend to cluster together (for example, pony club or breed associations) and group members may share similar risk-related beliefs. It may be that feral horse owners, vendors and purchasers as a group accept a higher degree of risk than domestic horse owners. In our study, multivariate analysis using a reduced model (Figure 4) further supports this observation. Estimated marginal means of the effect of the number of *warning/more work* terms on advertised price shows that an increased number of *warning/more work* behavioural descriptors led to a reduced price in domestic horses. This trend was not replicated in feral horses. With feral horses, as the number of *warning/more work* behavioural descriptors increased, the price did not alter. Our data appear to indicate that safety, rideability and training in feral horses may not be prioritised traits. Horse-related injury research in Australia differentiated by the breed of horse would unpick any unreported breed-related trends that may exist. 

Research has demonstrated the importance of achieving a good match between horses and riders [9,52] in the context of welfare for both parties. A ‘match’ or ‘mismatch’ horse-rider combination has been defined as the result of interactions between the dyad [9]. Horses which are mismatched with owners are more likely to experience compromised welfare [53]. They may be resold, resulting in increased levels of social instability, potentially multiple homes and training modalities [54]. Predictability is an important contributor to animal welfare [55,56] and as research has demonstrated horses’ ability to perceive certain human actions as threatening [57], the stability of a familiar, predictable owner must benefit the horses’ mental security. Security and predictability may be of increased importance for the welfare of rehomed feral horses as they adapt to domestic habitats. Most of the interventions experienced by domestic horses would be novel to the feral horse and many are in direct contrast to the equid ethogram [58]. These experiences include transport [59], separation from familiar conspecifics [60], training processes [61] and the novelty of situation and location [62]. Williams and Tabor (2017) recognised that in the human-horse dyad, it was the responsibility of the human to ensure the health and welfare of the domestic horse were optimised through appropriate management and riding [63]. Correspondingly, it has been acknowledged that, while the welfare impact of the rehoming and domestication process in the feral horse are largely unknown, impacts would be dependent on the skill, knowledge and training approach of the humans involved [4,64]. Despite its broad social appeal, the welfare impacts of rehoming and ensuing domestication are subject to many variables, as outlined here. All of these may potentially affect the ultimate success of the original rehoming event, subsequent on-selling and the corresponding welfare impact on each feral horse and its owner/s. 

An important consideration when determining the target market for feral horses is height. Feral horses were under-represented at taller heights with only ten (7.8%) advertised as over 15 hands. Indeed, 41.4% (*n* = 53) measured at less than 14 hh. The international governing body of equestrian sports (FEI) measuring system stipulates that horses measuring less than 14.2hh are classified as ponies [65]. Ponies (and smaller horses) are usually the horse of choice for junior riders given the smaller stature of both. We have referred to the recognised inherent risks of horse riding and handling in general [10,50,51] and these risks are magnified in the younger age bracket [66,67,68]. Hawson et al. (2010) recommend safety and suitability for purpose should be paramount when purchasing a horse or pony for a child [69]. Univariate analysis of our data showed feral horses were over-represented as *warning terms/needs work* descriptors exceeded two per advertisement compared with domestic-bred horses (Figure 4). This suggests that, despite their smaller size, feral horses may not be the optimal choice for junior riders due to safety concerns and, as smaller horses better suited to experienced or advanced riders, they may indeed suit a relatively narrow demographic.

The variables age and colour, which along with height are attributes intrinsic to the horse, were also significant, although neither the “age” or “colour” interaction terms reached significance in the multivariable mixed model (χ^2^ = 4.48, *df* = 4, *p* = 0.345 and χ^2^ = 4.27, *df* = 6, *p* = 0.640 respectively). Parks Victoria’s feral horse rehoming program requires submission of an expression of interest, including criteria that must be met before an application can be considered [70]. Applicants have no opportunity to specify which age, sex, colour etc. of feral horse they prefer to receive. In contrast, feral horses trapped by the National Parks and Wildlife Service (NPWS) in NSW must be sourced through a rehoming group or individual if fewer than five horses are required. If a member of the public wishes to rehome five or more feral horses directly from NPWS, they are similarly required to apply, but horse preferences (age, sex, colour) are considered [71]. Therefore, depending on the original source of the feral horse, rehomers may have the opportunity to select a horse which, at least aesthetically, they find suitable. 

Feral horses were over-represented among younger age groups i.e., less than six years, with only 4.7% (*n* = 6) aged over ten. This suggests that feral horses captured for rehoming may be younger animals or that rehomers may select feral horses aged under ten, if given the opportunity to choose. With a domestic horse, the latter strategy is reasonably common and typically has pros and cons. Among the pros: younger horses may have had fewer undesirable experiences with humans, be in better health and present the chance for an owner to “put their own stamp on them” than older horses. Conversely, older domestic horses typically have a learning history, and may have pre-existing health conditions or injuries, and it is generally accepted among equestrians that when you “buy a horse, you buy its history”. However, younger horses may require more work, exposure to novel events/experiences and training than older horses and therefore require an additional investment of time and/or money by the purchaser. As presented earlier, the importance of such requirements to purchasers varies according to the riders’ age, experience with horses and gender [47,48,49]. However, a feral horse, particularly one that hasn’t been sourced through a third party, does not necessarily fit this scenario. Neither younger nor older horses will have had any exposure or interactions with humans beyond the fear associated with the trapping event and subsequent removal from the trap-site. In this regard, feral horses of any age are essentially a “blank canvas” and every human intervention will be novel. As with domestic horses, it is possible that older feral horses are more likely to have pre-existing health conditions or injuries. The over-representation of advertised feral horses among younger age groups can essentially be attributed to three possibilities: (a) feral horses have a generally younger age distribution than the domestic population (wild animals tend to have shorter lifespans than their equivalents in captivity or domesticity [72]), (b) vendors have preferentially selected younger horses at rehoming and therefore have younger horses to resell or (c) vendors have chosen to keep older feral horses and resell their younger stock, possibly hoping for a higher sale price. Understanding the motives and preferences of purchasers and vendors of feral horses requires additional survey research to provide further insight into the target market for feral horses. 

Colour is an intrinsic trait that can add value to horses [73] and is of particular relevance to breeders who aim to produce foals with certain coat colours [74]. However, in the ridden horse, links between colour and rideability, behaviour, trainability and temperament are largely anecdotal with some studies identifying associations that are tenuous at best [75,76,77]. In our sample, coloured horses were under-represented among feral horses and dilutes were over-represented. As with age, this suggests that rehomers may preferentially select feral horses for colour, although with two colour categories (coloured and dilute) contributing to the difference between observed and expected values, the pattern is not quite as straightforward. Coloured feral horses were under-represented, so we are again presented with three possibilities that (a) vendors have chosen to keep coloured feral horses rather than sell them, (b) there are fewer coloured feral horses available for rehoming or (c) colour differences could reflect the colour distribution in the feral horses (or the captured proportion), regardless of their availability to the market. Conversely, dilutes were over-represented among feral horses, presenting two differing possibilities that either (a) vendors have preferentially selected dilute horses at rehoming and therefore more horses of this colour are resold, possibly for a higher sale price or (b) there are more dilute feral horses available for rehoming. Again, answers to these questions are outside the scope of the current study but are worthy of further investigation. Our modelling revealed a complex relationship among cost, ridden status and colour (χ^2^ = 51.38, *df* = 23, *p* = 0.001). Choosing a ridden horse based on colour is not unknown among equestrians and numerous anecdotal examples can be found in “horse wanted” advertisements. For example, it is not unusual for a “wanted” ad to specify “no chestnuts” or “no greys”. Whilst selecting a horse on aesthetics alone may have no safety impact with breeding animals, the practice in ridden horses is considerably more problematic. Horse-riding is a popular pastime and the majority of riders are amateurs and/or leisure riders [47]. Appropriate rider–horse matching and selecting for traits such as trainability, experience and temperament can help mitigate some of the risks in what is inherently a dangerous activity [10,11]. 

Our results suggest that Australian feral horses are valued differently from similar domestic bred horses, but questions remain about their utility in the recreational riding horse market. What niche, if any, is suitable for these horses and their purchasers to optimise horse-human safety and welfare? What are the intangible qualities that make a feral horse valuable, to vendors at least? It is worth noting that there has been considerable media attention and public scrutiny surrounding feral horse management in general, including rehoming [78,79]. Despite this publicity, it was surprising to find that feral horses (brumbies) comprised a tiny fraction of the horse market in the study period. We reviewed 15,404 photo advertisements of horses for sale in the “Allrounders”, “Coloured” and “$1000 and Under” sections in 53 editions of *Horse Deals* magazine, and found only 128 (0.0083%) were brumbies. The interest in rehoming horses from industries such as thoroughbred and harness racing has burgeoned in the last decade. Racing organisations have come under increasing scrutiny from the public to minimise horse wastage and support the transition of racehorses into new roles once their racing careers have ended. Industry heavyweights, including Racing Victoria, Racing NSW and the harness racing equivalents have created specific divisions to facilitate racehorse rehoming [80,81,82] supported by considerable capital investment. Indeed, *Horse Deals* magazine has a dedicated “Off The Track” section for vendors to advertise ex-racehorses and these have to be absorbed into an already saturated recreational riding-horse market [83]. Whilst brumbies differ vastly from thoroughbred or standardbred ex-racehorses, it is important to acknowledge that horse rehoming has become an extremely competitive market. Feral horse vendors would benefit from maximising their horses’ suitability to the target market to result in a successful outcome for all parties, including the horse. Perhaps most importantly, in the context of what is undoubtedly a complex, emotive issue, studies such as this can provide insights into the value and sustainability of rehoming as a feral horse management tool.

### Limitations

We recognise the vendors’ asking price may not reflect the ultimate sale price, but after-sale interviews with purchasers to determine the final sale price or the reason for the sale would have to be retrospective and were beyond the scope of this study. Raw data analysed in this study were collected from *Horse Deals* magazines from February 2017 to July 2022. Despite every attempt to source them, not every edition published during that period was available for analysis and 14 were missing. We acknowledge that advertisements could differ over time, and analysis of contemporary editions might yield different results. In some cases, the same horse might have been advertised in more than one issue, including missing issues. To account for this possible source of error, data for such advertisements were recorded only the first time the horse was advertised. 

There is also the possibility of error arising from our allocation of behavioural descriptors to categories. We used previously published categories [5,7,8] but added terms more relevant to the horses of interest i.e., horses rather than ponies. Additional terms were categorised after consultation between the authors, three of whom have relevant expertise as veterinarians, animal behaviourists and riding instructors. As noted in earlier literature [5,7,8], we acknowledge that the assignment of behavioural descriptors to categories is inevitably subjective, and one would expect disagreement among horse people. For example, the subtext of “good ground manners/good to handle/groundwork” could be “but not good to ride”; “good to ride in the open arena” could be “but not in an indoor/enclosed arena”; and “very sporty” might imply quick paced or a level of responsiveness that could intimidate a novice owner. Some advertisements seemed to contain contradictory statements, such as where a horse may be described as “quiet” but also described as “not for beginners”. In these cases, both statements were scored with +1 for each mention of a behaviour that corresponded with one of the four categories. Statements which described the vendors’ future expectations of the horse, such as “make good allrounder once broken in” and “potential barrel racer” were noted as aspirational in Table 2, as they did not reflect a current description of the horses’ behaviour. Some phrases in advertisements represent vendors’ interpretation of their horses’ behaviour [84]. These may reflect subjectivity and anthropomorphism, which should be considered when interpreting results.

## 5. Conclusions

We have demonstrated that Australian feral horses entering the riding horse market following capture and rehoming are valued differently to domestically bred horses of similar utility. The ultimate success of feral horse rehoming programs is largely dependent on the availability of suitable, knowledgeable homes to accept these horses. We question the suitability of feral horses for inexperienced or recreational riders as, based on our results, advertised feral horses generally appear to require significant training to become safe riding horses. However, this finding was not reflected by vendor prices in that as the number of *warning/more work* behavioural descriptors increased, the market price did not alter. The potential for horse/rider mismatch could result in a highly undesirable outcome for the horse and potentially the rider. 

## Figures and Tables

**Figure 1 animals-13-01481-f001:**
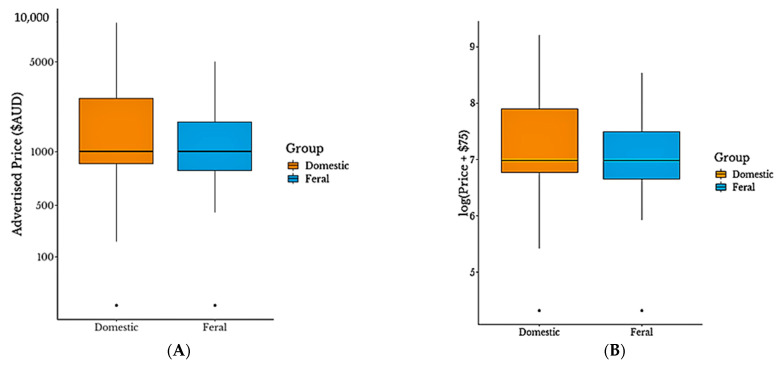
Distribution of (**A**) price (AUD) and (**B**) log(Price(AUD) + 75) of domestic horses (*n* = 256) and feral horses (*n* = 128) advertised for sale in the “Allrounders”(*n* = 6902), “Coloured”(*n* = 2383) and “$1000 and Under”(*n* = 6119) sections of *Horse Deals*, published February 2017 to July 2022. Datapoints more than 1.5 times the interquartile range from the median on the log scale are marked with a dot.

**Figure 2 animals-13-01481-f002:**
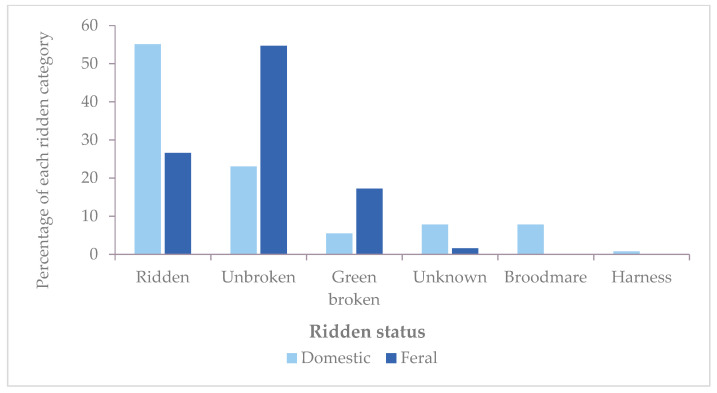
Distribution of the ridden status of domestic (*n* = 256) and feral (*n* = 128) horses in pooled advertisements (“Allrounders” (*n* = 6902), “Coloured”(*n* = 2383) and “$1000 and Under”(*n* = 6119) sections) in 53 editions of *Horse Deals* magazines, 2017–2022.

**Figure 3 animals-13-01481-f003:**
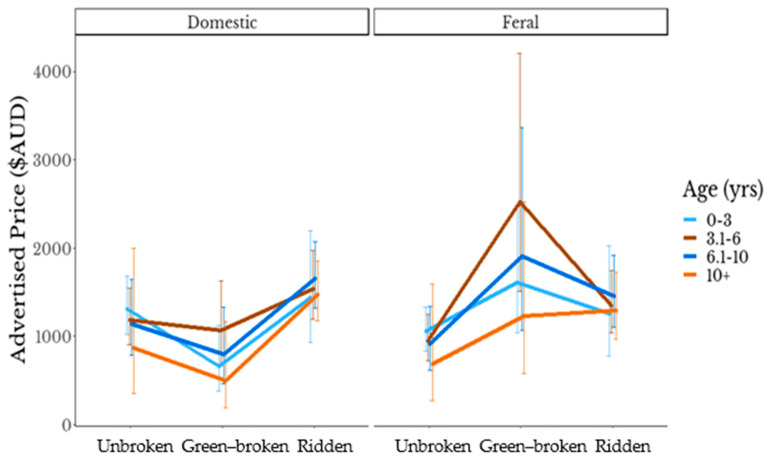
Estimated Marginal Means from the multiple linear regression showing the interaction of age and ridden status on advertised price for feral and domestic horses. Bars are 95% confidence intervals.

**Figure 4 animals-13-01481-f004:**
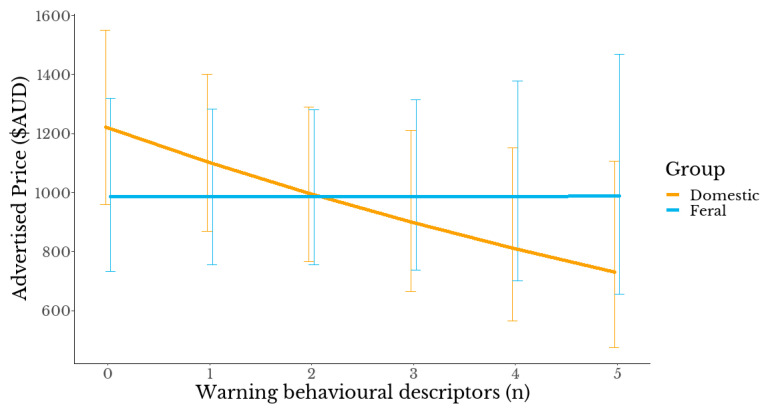
Estimated marginal means of the effect of the number of *warning/more work* terms on advertised price for domestic and feral horses using a reduced model. To facilitate the estimation of marginal means in the sparse dataset, a reduced model excluding the following interaction terms: (Ridden_status × Colour_inc_Dilute_category, Ridden_status × Age, Group × Ridden_status) was used. Bars are 95% confidence intervals.

**Table 1 animals-13-01481-t001:** Issues of Horse Deals magazine used to compile the dataset.

Year	Months
2017	Feb, Mar, Apr, Jun, Jul, Aug, Nov, Dec
2018	Jan, Feb, Mar, Apr, May, Jun, Jul, Aug, Oct, Nov
2019	Jan, Feb, Mar, Apr, May, Aug, Sep, Oct, Nov
2020	Feb, Mar, Apr, Jul, Aug, Sep, Oct, Nov, Dec
2021	Jan, Feb, Mar, Apr, Jun, Jul, Aug, Sep, Oct, Nov, Dec
2022	Jan, Feb, Mar, May, Jun, July

**Table 2 animals-13-01481-t002:** Behavioural descriptors allocated to the level of reassurance inferred by statements (*n* = 296) found in advertisements for feral and domestic horses.

Very Reassuring (*n* = 9 [3%])	Somewhat Reassuring (*n* = 52 [18%])	Neutral (*n* = 125 [42%])	Warning/More Work (*n* = 110 [37%])
Good to ride out alone	Very brave	Great work ethic	Some issues require knowledgeable home/some saddle issues
Calm/Relaxed/Laid back	More woah than go	Teachable/trainable	Cold-backed at first
Quiet	Great with games	One in a million	Energetic
Safe	Great with obstacles	Has dressage/on-the-flat training	Nervous/nervous with people
Unflappable		Safe/good with other horses	Anxious
Good on roads	Good to ride in an open arena/paddock	Taught to lead, tie, float	Can pigroot/buck
Suit/perfect first horse/pony	Honest, willing jumper/jumps confidently	Due to injury suit companion/light work/trail riding	Not a child’s pony
Trustworthy (with kids)	Good on trails	Cannot be ridden due to injury	May need professional to load/hard to load
Confidence builder	Good brakes/stop	Very keen	Wasting in paddock
	Almost bombproof	Sane	Lead line pony only
	Trained to harness	Goes over trot poles	Gelded late, needs owner who can handle a boy
	Champion led and ridden/placed at shows	Been to adult riding club and trail riding	Wild born
	Good/great with kids	Done station/stock work	Hard to catch
	Can be ridden by kids	Been to beach	Safe but not for nervous rider
	Suit a range of riders	Been to pony club	Safe/quiet but green
	Not spooky	Honest/kind eye	Quiet to ride but not for beginners
	Good to ride/under saddle	Ripper pony	Big movement so no timid riders
	Unraced/never raced	Always tries her hardest	Can catch with food/in yard
	Great family horse	Kid’s play pony	Can be headstrong
	Good in town	Good in a herd	Sold straight out of the paddock/as is
	Can be ridden in a halter/bitless	Good with fences	Nervy/nervous/stressy
	Never put a foot wrong/had any trouble with	Comes when called	Has separation anxiety
	Suit lady or novice	Easy to float/self-loads	Anxious with other horses, best off to be alone
	Does not get hot	Floats/floats ok	Anxious in new environments
	Perfect/good allrounder	Easy to catch/lead	Can spook
	Handled by children	Lovely on the lunge	Still learning to float
	Good ground manners/good to handle/groundwork	Can be ridden	Restless when tied up/needs work on tying up
	Sensible	Ridden by kids and adults	Quick on feet
	Good to trim/f/w/c	Easy load, truck or float	Can be nasty on the ground
	Gentle giant	Ready to compete	Can be stubborn to get going but great once going
	Easy to do anything with	Very sporty	Trusting once he knows you
	Great/good/beautiful temperament/nature	Suit broodmare	Standoffish
	No dirt/nasties	Broodmare or companion only	Mistreated in past and scared of strangers
	No vices	Kind	Can be lazy
	No fuss	Will be great riding club horse	Can get pushy
	Been there, done that/done the miles	Suit trails/PC/dressage	Experienced home/rider only
	No buck/kick/bolt/rear	Suit show/break in/breed/companion/led pony	Project
	Soft/snaffle mouth	Suit young rider to jump, do sports	Sensitive
	Willing/willing to please	Will suit any discipline	Not in work
	Well educated	Had confident beginners on in yard	Can be forward/forward moving
	Absolute gentleman	Needs firm, gentle approach	Girthy
	Good after a spell	Dressage/hack prospect	Cheeky
	Well-mannered/behaved	Suit western or pleasure Suit slow events	No kids/beginners
	Good alone or in company	Broodmare/proven broodmare	Needs confident/competent rider
	Does not need work	Throws flashy colours/nice foals	Timid/shy
	Suit lead rein/been lead rein pony	Great mum	Ready to go on with
	Taught children to ride	Loves attention	Suit intermediate rider
	Well-mannered under saddle and on ground	Used as a companion	Restarted/reschooled
	Very easy horse	Suit companion	Suit confident, older teen/adult
	Suit anyone	Owner has no time	Saved as orphan/hand reared
	Good manners/well mannered	Clean slate	Can hand feed
	Professionally broken in/trained	Follows you/wants to be with you	One owner type
	Good with dogs/cars/machinery	Lovely	Currently yarded but no halter on
		Sweet	Needs a lunge before riding
		Curious/inquisitive	Need to be gone asap
		Beautiful	Needs knowledgeable home
		Generous	Needs confident/loving handler
		Friendly	Needs patient/experienced handler/rider
		Loving	Needs groundwork
		Great personality	Needs time and education
		Lovable	Needs truck/stock crate to transport
		Fun	Needs a job
		Competitive	Needs balanced, confident, sympathetic rider
		Wonderful horse	Needs work
		Good boy	Needs active rider
		Good breeding/well bred	Needs good fences/electric
		Ex-pacer/trotter	Needs regular work/a few times a week
		Station bred	Captured/trapped as part of capturing program ‡
		Holds weight well	Domesticated for over a year ‡
		Loves jumping	Very quiet for a brumby ‡
		Limited jumping	Makes great allrounder once started/broken in *
		Done trail riding	Make great teens horse once broken *
		Affectionate/cuddly	Make an excellent breaker if someone had time *
		Sure-footed	Good horse/will excel, in right hands *
		Playful	Ex-racehorse †
		Look at my presence	Off the track 3/4 weeks etc. †
		Grown out naturally	Off the track for a year †
		Not the best paddock mate	Ready to retrain †
		Low maintenance	Needs finishing/unfinished †
		Good nature when given attention	Learning to tie †
		Not suited to racing	Halter broken †
		Unsound/not sound to ride	Going kindly in walk and trot †
		Food aggressive but not to people	Recently started under saddle †
		Very smart/intelligent/quick learner	Green/new to saddle †
		Sound	Basic education †
		Waits at gate to be ridden	Has basics †
		Trained using Parelli/Natural Horsemanship	Blank canvas †
		Go all day	Needs more work/flat training/education †
		Picks up feet and will load on float	Unbroken †
		Been ridden/sat on bareback	Green broken †
		Ready for new adventure	Broken in †
		Good to trim/easy to do feet/can trim	Has had saddle on/been sat on †
		Good/great movement/paces	Started and turned out †
		Never foundered	Had 2/a few/a dozen rides †
		Good home only	Ready to start under saddle/break in †
		Quiet but reactive	Been saddled/long reined/mouthed †
		Has been to a few competitions	Needs a lot of work †
		Previously ridden by children	Sold as unbroken †
		Responsive	Basic groundwork †
		Ready for forever home	Unhandled †
		Just needs love	Lightly handled †
		Good doer/easy keeper	Basic handling/had some handling †
		Suit riding or harness	Needs lots of handling/not had much handling †
		Heart of gold	Leads/only broken to lead/learning to lead †
		Well socialised	Been ridden a few/a handful of times †
		Awesome	Lightly ridden †
		Not colty at all	Lightly lunged and backed †
		Walks through forest	Needs someone to bring back into work/continue riding †
		Focused	Needs to be rebroken †
		Potential barrel racer/cowhorse *	Ties and leads †
		Make excellent allrounder/PC/ARC *	
		Make good campdraft/mustering horse *	
		Make great show pony *	
		Make great showjumper/sporthorse *	
		Make lovely kid’s pony *	
		Make beautiful pony club horse *	
		Make good trail/pack horse/allrounder *	
		Make perfect mount to ride/show/compete *	
		Will be float trained on sale *	
		Future eventer *	
		Future kids pony *	
		Has (great) potential/big future *	
		Make good allrounder once broken in *	
		Make loyal riding horse *	

* Aspirational statement. † More work required before riding as a recreational mount. ‡ Specific to brumbies.

**Table 3 animals-13-01481-t003:** Interaction terms among the fixed effects with *p* < 0.25. These were simultaneously added to the reduced model to produce the final model.

Term	χ^2^	*Df*	*p* Value
Ridden_Status × Colour_inc_Dilute_category	41.8596	23	0.009426
Age × Somewhat_reassuring	10.4854	4	0.03300
Group × Somewhat_reassuring	3.9398	1	0.047158
Group × Warning_Needs.Work	3.6355	1	0.05656
Ridden_Status × Age	25.2038	16	0.066312
Group × Colour_inc_Dilute_category	11.3812	6	0.077286
Group × Ridden_Status	5.0278	3	0.169773
Somewhat_reassuring × Sex_A	3.0224	2	0.220645

**Table 4 animals-13-01481-t004:** Explanatory variables reaching significance by Chi-square or Fisher’s exact test. Expected values from the null hypothesis appear in brackets.

		Count		
Explanatory Variable	Category	Domestic	Feral	*p* Value
Ridden status	Broodmare	20 (13.33)	0 (6.67)	Fisher’s exact test
	Green_broken	14 (24.00)	22 (12.00)	*p* < 0.001
	Harness	2 (1.33)	0 (0.67)	
	Ridden	141 (116.67)	34 (58.33)	
	Unbroken	59 (86.00)	70 (43.00)	
	Unknown	20 (14.67)	2 (7.33)	
Height	<14 hh	61 (76.00)	53 (38.00)	Fisher’s exact test
	14 – <15 hh	52 (76.67)	63 (38.33)	*p* < 0.001
	15 – <16 hh	87 (64.00)	9 (32.00)	
	16 + hh	54 (36.67)	1 (18.33)	
	not listed	2 (2.67)	2 (1.33)	
Age (years)	3 yrs or younger	49 (68.67)	54 (34.33)	Fisher’s exact test
	3.1 to 6 yrs	61 (70.67)	45 (35.33)	*p* < 0.001
	6.1 to <10 yrs	70 (61.33)	22 (26.00)	
	>10 yrs	72 (52.00)	6 (26.00)	
	unknown	4	1	
Colour	Bay	86 (91.33)	51 (45.67)	Chi-square test
	Black	14 (13.33)	6 (6.67)	*p* = 0.008
	Brown	20 (20.67)	11 (10.33)	
	Chestnut	54 (57.33)	32 (28.67)	
	Coloured	47 (35.33)	6 (17.67)	
	Dilute	17 (22.00)	16 (11.00)	
	Grey	18 (16.00)	6 (8.00)	
Warning terms/needs work				
	0	90 (76.67)	25 (38.33)	Fisher’s exact test
	1	99 (92.67)	40 (46.33)	*p* < 0.001
	2	46 (58.00)	41 (29.00)	
	≥3	21 (28.67)	22 (14.33)	

**Table 5 animals-13-01481-t005:** Results of the multivariable regression between price and explanatory variables of domestic horses (*n* = 256) and feral horses (*n* = 128) advertised for sale in the “Allrounders” (*n* = 6902), “Coloured” (*n* = 2383) and “$1000 and Under”(*n* = 6119) sections of *Horse Deals*, published February 2017 to July 2022.

	χ^2^	*Df*	*p* Value
(Intercept)	2719.60	1	<0.0001
Group	5.64	1	0.018
Ridden_status	23.43	5	<0.0001
Age_cat	4.48	4	0.345
Colour_inc_Dilute_category	4.27	6	0.640
Very_reassuring	2.53	1	0.112
Somewhat_reassuring	0.79	1	0.375
Warning_Needs.work	7.29	1	0.007
Group × Ridden_status	14.31	3	0.003
Group × Warning_Needs.work	1.47	1	0.225
Ridden_status × Age_cat	26.04	16	0.053
Ridden_status × Colour_inc_Dilute_category	51.38	23	0.001
Age_cat × Somewhat_reassuring	7.69	3	0.053

## Data Availability

Restrictions apply to the availability of these data. Data was obtained from *Horse Deals* (Australia) and are available from the authors with the permission of *Horse Deals* (Australia).

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
