# Peer review of "Investigating the Market Value of Brumbies (*Equus caballus*) in the Australian Riding Horse Market"

_animals, 2023, doi:10.3390/ani13091481_

Round 1
Reviewer 1 Report
This paper assessed the selling price of presumed feral horses in 128 advertisements and domestic horses from 256 advertisements. The advertisements were assessed for phrases relevant to behavioral characteristics, as well as physical characteristics. Analyses centered around how these factors differed between feral and domestic horses, and how they affected price of the horse. The authors found that feral horses for sale tended to be younger than domestic horses, and were generally cheaper. There was less of an effect of how trained a feral horse was on its selling price than for domestic horses.
Feral horses are highly contentious in the USA and Australia, with rehoming the animals often touted as a means to solve the issue of their overpopulation. This research is important both to show the perceived value of feral horses in the marketplace, and to show how commonly (or not) they are sold on. It was interesting, although unsurprising, that they are considered cheaper than domestic horses. Details revealed in this paper are useful for managers of horse populations, and for those who may be considering acquiring a feral horse with the prospect of selling it. An unknowable that is not addressed in this paper is the reason why these animals are being sold. It would be an interesting discussion point to assess that maybe these horses are being sold because of the issues in the warning/more work category, whereas horses in the reassuring categories are more likely to be kept. The paper should also address other potential fates for feral horses (i.e., whether they can also be sold for slaughter).
Some of the valence assigned to descriptors could be disputed – it was not clear why some of the terms assigned to neutral that seemed very complimentary were not put in one of the reassuring categories. More description about how many people reviewed these terms and made these choices, what experience they had, and how the choice was made would be useful.
The results section should be re-written to include the statistical results where the variate is described, rather than having a section for statistical results. All figures should be re-done to remove the grey background, increase font and line size, and make the color scheme easier to read.
The discussion is over-long and should be shortened. There is quite a bit of repetition and extraneous detail.
L46-48 – this makes it appear that the mean price of a brumby was actually more than a domestic horse which contradicts L39-40. It is not clear what point is being made here if brumbies are in face cheaper, so would benefit from rephrasing.
L57 – Kaimanawa horses are from a specific population. There is another population in NZ, so it might be worth checking what the general term is in that country.
L94-95 – remove ‘Of course’.
L113-114 – so is there currently no antagonism from pastoralists? This would be surprising.
L167 – change “priced” to “price”.
Table 2 – “Gentle giant” is repeated. Repeated column headings should be removed.
L227 – why was rider experience dismissed? Experience of the rider may lead to how a horse is portrayed in the advertisement.
Figure 1 – explain what the dots are in the figure. Also it would be good to make the axis labels bigger, to remove the grey background, and to change the color scheme to be more accessible to the colorblind.
L249 – why was advertisement size included? Surely this is as much to do with wealth of the advertiser as value of the horse?
L245-263 – sample size seems small for this kind of analyses.
L278 – price is not shown in table 4. Remove this sentence as it is explained on L283.
L280-281 – I fail to see the relevance of the collective price and metrics.
L318 – these results should be added to the relevant section where the variate is described rather than lumped together in an analysis section.
Figure 3 is hard to read. It is not showing the interaction just presenting data of the 3 factors.
L399-402 – this is repetition and can be removed.
L460 – the authors need to consider that brumbies may be smaller than the average domestic horse as they have undergone some natural selection, and/or may be a product of smaller breeds.
L503-504 – this is an assumption that needs a citation or should be removed. Bait trapping is typically considered low stress, but any handling of a feral animal will be stressful.
Reviewer 2 Report
I really enjoyed reading your paper. It was very well written; the methodology and results were easy to follow and understand, and overall, it provided a fascinating insight into the market value of brumbies. Great work by all!
Some (very) minor comments below:
Line 159 Horse Deals
Line 166 Why the choice of Guy Fawkes as a breed descriptor (for those of us less familiar with brumbies)?
Line 210 What does registration status relate to? I’m not sure what you are meaning they are registered or not registered with
Line 447 This reference may be relevant here Liehrmann, O., Viitanen, A., Riihonen, V., Alander, E., Koski, S. E., Lummaa, V., & Lansade, L. (2022). Multiple handlers, several owner changes and short relationship lengths affect horses’ responses to novel object tests. Applied Animal Behaviour Science, 254, 105709.
